# Migration of Microplastics and Phthalates from Face Masks to Water

**DOI:** 10.3390/molecules27206859

**Published:** 2022-10-13

**Authors:** Giuseppina Zuri, Bernat Oró-Nolla, Ana Torres-Agulló, Angeliki Karanasiou, Silvia Lacorte

**Affiliations:** Institute of Environmental Assessment and Water Research of the Spanish Research Council (IDAEA-CSIC), Jordi Girona 18-26, 08034 Barcelona, Spain; gzuqam@cid.csic.es (G.Z.); bonqam@cid.csic.es (B.O.-N.); ana.torres@idaea.csic.es (A.T.-A.); angeliki.karanasiou@idaea.csic.es (A.K.)

**Keywords:** face mask, plastic polymers, microplastic, phthalate, plasticizers, migration

## Abstract

Since the outbreak of COVID-19, face masks have been introduced in the complex strategy of infection prevention and control. Face masks consist of plastic polymers and additives such as phthalates. The aim of this study was to evaluate the migration of microplastics (MP) and phthalates from face masks to water. Four types of masks including FFP2 masks and surgical were studied. Masks were first characterized to determine the different layers and the material used for their fabrication. Then, masks were cut into 20 pieces of 0.5 cm^2^, including all their layers, placed in water, and the migration of MP and phthalates was evaluated according to the conditions stated in EU Regulation No 10/2011 on plastic materials and articles intended to come into contact with food. For MP, the morphological analysis (shape, dimension, particle count) was performed using a stereomicroscope, while the identification of both masks and MP released was conducted using μ-Fourier-transform infrared spectroscopy (µ-FT-IR). Migration of phthalates was assessed by ultra-high-performance liquid chromatography coupled to triple quadrupole mass spectrometer (UPLC-MS/MS). Face masks analyzed in the present study were made of atactic polypropylene (PP) as stated by the manufacturer. The μ-FT-IR confirmed that PP and polyamide (PA) were released as fragments, while both PP and polyester (PES) were released as fibers. In addition, 4 phthalates were identified at concentrations between 2.34 and 21.0 µg/mask. This study shows that the migration study can be applied to evaluate the potential release of MP and phthalates from face masks to water and could give a hint for the potential impact of their incorrect disposal on the aquatic resources.

## 1. Introduction

To reduce and prevent the transmission of the SARS-CoV-19 pandemic, governments all over the world were urged to introduce the use of protective equipment such as face masks [1]. Different kinds of masks are being used: surgical or medical, FFP2, N95, cotton, fashion or activated carbon masks. Variations in the manufacture of face masks exist and depend on the manufacturer. Surgical masks consist of three layers: an outer layer (nonwoven fibers), a middle layer (melt-blown fiber), and an inner layer (soft fibers) [2]. FFP2 masks consist of 5 layers: 2 layers outer of non-woven fabric, 2 layers of melt-blown fiber, and 1 inner layer of soft fibers. Each layer exerts a specific function: the outer layer effectively restrains and isolates large particles and dust; the middle layer is the main filtering layer; the inner layer is hypoallergenic and permeable to air, making the mask comfortable to wear. Their manufacturing materials consist mostly of polymers with polypropylene (PP) being the most widely used [3], although polyethylene (PE), polyamide (PA), and polyethylene terephthalate (PET), or polyester (PES) as commonly termed in textiles, can be used [4]. Different masks contain different amounts of plastic polymers: surgical masks contain approximately 4.5 g of PP, while FFP2 contain about 9 g [5]. 

An incorrect disposal of masks can introduce a massive amount of plastic into the environment [6,7]. Different processes such as UV radiation, mechanical stress or hydrolysis might break down larger plastic pieces into microplastics (MP) [8]. A shear test was conducted to mimic mask degradation once disposed, resulting in the release of mean of 2.1 ± 1.4 · 10^11^ pieces/m^2^ of mask [9]. A leaching experiment showed that the number of MP and nanoplastics (NP) release by a single face mask in water ranged from 1300 to 4400 MP/mask [10]. In addition, laundering of disposable and reusable face masks can represent a source of synthetic and natural microfibers to freshwater. It was observed that a single cycle of laundering in a domestic washing machine led to the release of a mean (±SE) of 284 ± 74 microfibers independently of the fabric composing the face mask [11], an amount that is not negligible considering the daily use of masks. Due to the unprecedented demand of disposable masks (about 89 million pieces per month) [1], the release of MP and chemicals into the aquatic environment could be significant upon improper disposal. 

Face masks also contain plasticizers [12], a group of low-molecular-weight substances, added to the polymer manufacturing process to enhance their flexibility and versatility [13]. Some plasticizers have been classified as potentially harmful for the environment [14] and human health [15]. The most widely used are phthalates, esters of ortho-phthalic acid, and organophosphate esters (OPE) [16]. The concerns related to phthalates are attributed to their ability to migrate from the plastic material over time, due to the absence of covalent bonds with the polymeric matrix. They are then released into the environment, specifically to surface water, air, and sediments, and can bioaccumulate in biota [17]. In humans, they are associated with premature delivery in pregnant women [18] and increased risk of diabetes [19], as well as producing carcinogenic, teratogenic, and mutagenic effects and acting as endocrine and metabolic disruptors [20]. 

This experimental study evaluates the migration of MP and phthalates from face masks to water to estimate the amount released to the aquatic resources upon disposal. The procedure applied is based on Regulation No 10/2011 used to determine plasticizers in food packaging and allows simulating the fate of mask components to the aquatic environment under realistic conditions [21]. We compared the results herein obtained with the few available in the open bibliography to evaluate the risk of improper mask disposal to the aquatic environment. 

## 2. Materials and Methods

### 2.1. Chemicals and Reagents

Phthalate Esters–Mix containing DMP (dimethyl phthalate), DEP (diethyl phthalate), BBP (butylbenzyl phthalate), DBP (dibutyl phthalate), DPP (dipropyl phthalate), BMPP (bis(4-methyl-2-pentyl) phthalate), DNHP (di-hexyl phthalate), HEHP (hexyl-2-ethylhexyl phthalate), DEHP (diethylhexyl phthalate), DNOP (di-n-octy phthalate) and DNP (di-n-nonyl phthalate) was purchased as a stock solution at 1000 μg/mL from LGC Standards (Augsburg, Germany). Triphenyl phosphate d-15 (TPhP d15) was used as internal standard and acquired from Sigma-Aldrich (St. Louis, MO, USA). 

Water, and organic solvents dichloromethane (DCM), hexane (Hex), ethyl acetate (EtAc), and methanol (MeOH) were used for the extraction and analysis and were acquired from Merck (Darmstadt, Germany). Formic acid for the mobile phase was purchased from Panreac, (Barcelona, Spain). We used 96% ethanol (Honeywell, Charlotte, NC, USA) to rinse the material. Water was filtered on 25 mm Nylon filters 20 μm ∅ (Merck Millipore, Carrigtwohill, Ireland). Samples were stored in glass Petri dishes (Brand^®^, Wertheim, Germany).

### 2.2. Mask Types and Characterization

Four masks of different models, brand, color, and manufacturing materials were studied (Table 1). Surgical and FFP2 masks were used as representative samples since they were the recommended masks to wear and the most widely used during the pandemic. Surgical masks were provided to IDAEA-CSIC personnel during the pandemic. Each mask was cut into a piece of 0.5 cm^2^ for characterization to identify the type of polymer for each layer. Characterization was performed with Fourier-transform infrared (FT-IR) spectroscopy (Thermo Avatar Nicolet 360, Thermo Electron Inc., Waltham, MA, USA) with a deuterated triglycine sulfate (DGTS) detector using the following setup: spectral range 525–8000 cm^−1^, 32 scans per sample, resolution of 4 cm^−1^. Spectra obtained were compared with the Hummer Polymer library and Nicolet Standard Collection Spectra. 

### 2.3. Migration Experiment

Migration experiments were performed according to the European Union “Commission Regulation (EU) No 10/2011 of 14 January 2011 on plastic materials and articles intended to come into contact with food” since these conditions mimic the realistic scenario in which MP and phthalates could potentially migrate from the mask once released in the aquatic environment. Masks were cut in square pieces of 0.5 cm^2^ (0.005 dm^2^) (including all layers), each weighting 0.004 g approximately, and 20 pieces were placed in Pyrex tubes that were filled up with 20 mL of HPLC water. Afterwards, vials were placed on an Orbital Shaker and constantly shaken for 10 days at room temperature (approximately 20 °C). Subsequently, the mask squares were removed from the vials using tweezers. The water content was filtered through 20 µm nylon filters of 25 mm ∅ and filters were analyzed for MP, while water was extracted to determine the release of phthalates. 

To minimize external contribution of both MP and phthalates, non-volumetric materials were rinsed 3 times with deionized water and placed in an ultrasonic bath with a small amount of strong base (Extran, Merck, Darmstadt, Germany) for 30 min. Afterwards the material was rinsed again with deionized water, and then with acetone, wrapped with aluminum foil and muffled in an oven at 400 °C for 8 h to remove residual organic pollutants. Glass Petri dishes and tweezers were rinsed with EtOH:H_2_O 70:30 (*v*/*v*). 

### 2.4. Microplastics Analysis

Morphological analysis of particles released from masks in the migration experiment was carried out using a stereomicroscope (Leica S9i, Wetzlar, Germany) equipped with a 10 MP integrated digital camera that provides full HD-microscope images. It consisted on determining the abundance and length (fibers), diameter and area (fragments). Particles retained on the filter were collected using Precision Tweezers (Dumont, Montignez, Switzerland) and placed on a CaF_2_ support. The identification of MP was achieved using a μ-FT-IR (Thermo Nicolet iN10 MX/Omnic version 7.3, Waltham, MA, USA) equipped with a mercury cadmium telluride (MCT) array imaging detector in transmission mode with spectra comparison with Hummel Polymer Sample Library. The spectral range was set from 800 to 4000 cm^−1^, performing 64 scans per sample at a resolution of 4 cm^−1^. To control background external contamination, 3 blank samples (water without masks) were analyzed using the same extraction protocol as for the mask samples. Concentrations are given in MP/dm^2^ and MP/mask where the amount detected in the 20 pieces of 0.5 cm^2^ used to perform the migration test was extrapolated to dm^2^ or converted into MP per mask considering the surface area of each mask in order to be able to compare the results obtained with other studies.

### 2.5. Phthalate Analysis

The water from the migration experiment was spiked with 0.5 µg of internal standard (IS). Phthalates were extracted from water samples from the migration experiment using a liquid–liquid extraction with three organic solvents with different polarities added individually with the following sequence: 10 mL of Hex, 10 mL of DCM, 10 mL of EtAc. Samples were vortexed for 1 min, sonicated in an ultrasonic bath for 1 min and centrifuged at 3500 rpm for 10 min. Then, supernatants from each solvent were pooled and evaporated with a TurboVap^®^ LV Concentration Evaporation Workstation from Biotage (Uppsala, Sweden) under a gentle N_2_ stream to near dryness. All the extracts were reconstituted with 500 μL of MeOH and placed in amber chromatographic vials and stored at −21 °C until analysis. 

Analysis was performed by ultra-high-performance liquid chromatography coupled to a triple quadrupole mass spectrometer (UPLC-MS/MS) (XEVO TQS from Waters Corporation, Milford, MA, USA). Electrospray ionization was set to positive mode (ESI+). Chromatographic separation was achieved by an Acquity BEH C18 (100 mm × 2.1 mm ID × 1.7 μm) with an Acquity BEH C18 VanGuard pre column (5 mm × 2.1 mm ID × 1.7 μm), both from Waters. A volume of 10 μL was injected and mobile phases consisted of (A) MeOH and (B) HPLC grade water, both solutions with 0.1% of formic acid. Elution gradient was from 40% A to 100 A in 8 min (4 min hold time) and to initial conditions in 2 min. Flow rate was set to 0.3 mL/min and the column temperature was set at 40 °C. MassLynx^TM^ 4.1 and TargetLynx^TM^ 4.1 from Waters were used for sequence programming and data processing. Acquisition was performed in Multiple Reaction Monitoring (MRM) using 2 transitions per compound. The ionization conditions were optimized from 10 to 60 V and the voltage that provided the most intense [M + H]^+^ ion was selected. MRM was then optimized from 10 to 100 eV to obtain the most 2 intense fragment ions. Two MRM transitions were monitored: the first one for quantitation and the second one for confirmation. The conditions used are indicated in Table 2. 

Calibration was done at 0.025, 0.050, 0.075, 0.25, 0.5, 0.75, 1 μg/mL, and each calibration solution contained 0.5 μg/mL of internal standard. The repeatability was analyzed by injecting the standard solution at 0.25 µg/mL in 5 consecutive times and the reproducibility by injecting the same solution in 5 different days. Extraction efficiency was performed using 20 mL of HPLC water fortified with 0.5 μg of analytical standards and 0.5 μg of internal standard, in quintuplicate (concentration of 0.025 μg/mL). IDL were calculated using a signal/noise ratio (S/N ratio) of 3 from the 0.025 μg/mL standard. Finally, blank samples (*n* = 5) consisted of HPLC water spiked with 0.5 μg of the internal standard, and the contribution of the different compounds detected was subtracted in the final calculations. Quality parameters of the analytical method developed are indicated in Table 2. Concentrations are given both as mg/dm^2^, according to Commission Regulation (EU) No 10/2011, annex 17 [21] and µg/mask. 

## 3. Results and Discussion

### 3.1. Characterization of Masks

Facial masks were disassembled into the different layers. FFP2 masks consisted of five layers, while the surgical mask only had three. According to the identification study performed with the FT-IR, all the masks were made of atactic PP. The matches between the spectra obtained from the analysis and the PP spectrum present in the library ranged from 95.58% for FFP2 B to 96.75% for FFP2 N (Table 1). Figure 1, Figure 2, Figure 3 and Figure 4 show pictures and structure of each mask analyzed. 

### 3.2. MP in Masks from Migration Experiment

Data from the migration study prove that all the masks considered in this study released particles in form of fibers and fragments, and their concentration is reported in Table 3. Table 3 also indicates the MP blank contribution of HPLC grade water used in the migration experiments. In two blank samples PA fragments were identified and in the other blank a PES fiber was detected, as well as cellulose fibers. This contribution was subtracted from the mask samples. Figure 5, Figure 6 and Figure 7 show the fibers detected in blank samples. 

Fibers migrated from the masks to a higher extent than fragments (Table 3), and this was probably linked to the fact that clothing material tends to release particles in form of fibers [22]. The fragment’s fraction released by face masks consisted of PP with 0–25% of total fragments, followed by PA fragments that were detected in three masks in percentages from 21 to 33% of total fragments. Two plastic polymers detected in form of fibers were PP and PES, the former accounting for 9 (FFP2 N) to 23% (FFP2 A) of the total fibers released, while PES accounted for 4 (Surgical Mask Q) to 26% (FFP2 N) of total fibers. Cellulose accounted for 50% to 79% of total fragments and from 65% to 85% of total fibers, and was the main non-synthetic polymer in all studied masks, associated with the material used in the inner layers. 

No trend was observed in the release of MP in form of fibers and fragments from different face masks depending on their number of layers. In fact, although FFP2 A and FFP2 B released 25% of MP in form of fragments, FFP2 N and Surgical Mask 2 released PA as fragments. The highest percentage of MP was released in form of fibers, and slight differences were observed among brands (Table 3). 

Table 4 indicates the dimensions of fragments and fibers released from face masks. MP detected had approximately 100–300 µm length, and larger fragments were found in FFP2 B and in the surgical mask. Fibers detected were in the range of 200-460 µm, with surgical masks with longer fibers. Also, fibers in the blank samples were much larger than in the samples. Figure 8, Figure 9, Figure 10 and Figure 11 show some of the fibers and fragments characterized in each face mask during the morphological analysis.

Since the FFP masks measure 2.48–2.55 dm^2^ and the surgical mask area measures 2.62 dm^2^ (Table 1), we estimated that each mask could potentially release from 2040 to 4716 MP/mask (Table 5). An overview of the results obtained in the present study and the literature concerning the release of MP from face masks is also presented in Table 5. Our results are comparable to those obtained by a migration study of MP from face masks in water performed by Shen et al. to assess the release of MP after single mask reuse [23] and a leaching study performed by Ma et al. [10]. In contrast, the results obtained in the present study are 10 times higher when compared with breathing simulation [24] and much lower when compared to studies of MP release involving mechanical or UV degradation of face masks [9,25]. 

A growing volume of MP is found in the environment, including the marine environment, freshwater, and atmospheric fallout [26]. According to the literature, MP in rivers differs up to 6 orders of magnitude, ranging from 0.0004 particles/L in the Austrian Danube [27] to 187 particles/L (mean 100 ± 49 particles/L) in surface water samples collected in the city of Amsterdam [28]. Wastewater treatment plants have been identified as a direct source of MP to rivers, although the discharge of plastic to rivers release MP due to fragmentation and weathering of plastic material. This study, similar to previous studies on the migration of MP from masks, indicates that masks can represent an additional source of MP to rivers, especially in cases where masks reach surface waters intact. Considering that thousands of MP are released to water from each mask (Table 5), and considering the huge amount of masks used during the COVID-19 pandemic, the MP released to the environment can represent a problem in large urban areas. As mask disposal is not regulated, it is estimated that around 50–70% of masks will reach landfills together with general waste, which can pose a problem due to migration of MP or plasticizers to the leachates. Another fraction will be incinerated. A part will be lost during transport or waste management. Another fraction will be directly released to continental or marine waters, depending on the waste management of each country. Considering the mean MP concentration released from the four masks studied, and considering that only 10% of the used masks reach surface waters and that 1 mask is used per person every 2 days as average, Figure 12 shows the gross amount of each MP polymer that will be released to waters per person per year. Fibers and fragments of PP, the main component of MP in face masks, is the main polymer that will be released to waters, followed by PES and PA. Cellulose, despite not being a synthetic MP, will also be released at high proportions. The problem can be serious in big cities or megacities, where waste disposal is not fully controlled. UNICEF has already highlighted the need to dispose of masks properly to avoid a serious future environmental problem [29]. 

It has been suggested that MP affect biota [10] as they can bioaccumulate in the food chain, and also represent a health hazard for humans through the ingestion of contaminated food or water [2,9,10]. Despite the knowledge gap concerning the human exposure to MP and health effects, in vitro toxicity studies highlighted the ability of MP to trigger immune response [30,31]. Additionally, MP could carry other potentially harmful compounds and heavy metals [32,33] that can be introduced in the human body.

### 3.3. Phthalates Released from Masks

In the samples analyzed, four phthalates have been systematically identified in the four masks studied: DBP, BBP, DNOP, and DEHP. The migration of these phthalates from the masks implies that they are mask components used during fabrication. Figure 13 shows the concentrations detected per mask in the migration experiment. The total concentration was 35 µg/mask for FFP2 A, 22.4 µg/mask for FFP2 B, 23.4 µg/mask for FFP2 N, and 25.3 µg/mask for the surgical mask. The amount was quite similar among masks and not associated with the number of layers constituting the mask. 

DBP was the main phthalate detected in FFP masks, at concentrations between 2.56 and 21 µg/mask. DBP is used as a plasticizer for vinyl fabrics, food wraps, curtains and raincoats [34,35], although ECHA has no information regarding all the registered uses. BBP was the second most abundant phthalate detected in masks at concentrations from 5.6 to 13.7 µg/mask, with a higher proportion in Surgical Mask Q compared to FFP2. The primary use for BBP is as a plasticizer in adhesives and sealants, floor coverings, paints, and coatings, and in plastic and rubber products [35]. DNOP was also detected in all masks at concentrations from 3.95 to 5.07 µg/mask. DNOP is one of the most widely used plasticizers worldwide due to its low cost, good stability to heat and UV-light, although it has been banned in many applications in the United States and Europe, unlike Asia [36]. Finally, DEHP, a common phthalate used in multiple applications, especially as an additive in polyvinylchloride plastic, was the least detected compound, at concentrations between 2.34 and 4.54 µg/mask. DEHP is used in the production of materials used for construction, adhesives, paint, and plasticizers [37]. The masks taken into account in the present study were purchased from different manufacturers and therefore their composition differed in terms of plasticizers released. Considering the amount of phthalates detected in the four masks studied and the weight of each mask, the amount migrating represents between 0.00005 and 0.0005%, which is a very low amount but indicates a realistic release to water simulating environmental conditions. 

Very few studies report the migration of phthalates from face masks to water or other simulants. Table 6 compares the concentration of several phthalates released from different commercial face masks (FFP2, surgical, N95, and active charcoal) in mg/dm^2^, according to the open bibliography. The results obtained in the present study are 20 to 100 lower compared previous studies on face masks [38,39] or in the same order of magnitude as Xie et al. [12]. However, the large differences could be linked to the phthalate extraction process from face masks. While in this study, the migration was carried out in water as simulant, and subsequently phthalates were thereafter extracted from water with organic solvents, in other studies face masks were directly immersed in organic solvents to extract phthalates, thus increasing the extraction efficiency. In the present study, the most frequently detected phthalates were DBP followed by BBP, DNOP, and DEHP. A different trend was observed by Min et al., who found DEHP and BBP being the dominant phthalates in N95 masks, while DHXP (dihexyl phthalate) was followed by DEHP in the surgical mask [38]. Wang et al. found comparable results for N95, while they found DnBP (di-n-butyl phthalate) and DEHP being the dominant phthalates in surgical masks [40]. Considering all studies in Table 6, DEHP, DEP, DBP, and BPP were often detected but at differing concentration, indicating that composition of masks varies depending on the brand. Other phthalates have been studied in face masks such as diallyl phthalate (DAP), dihexyl phthalate (DHXP), bis(2- ethylhexyl)phthalate (DPhP), bis(2-n-butoxyethyl) phthalate (DBEP); didecyl phthalate (DDP); diisononyl phthalate (DiNP), di-cyclohexyl phthalate (DCHP), diisobutyl phthalate (DiBP), and DnBP, although few data exist [12,38,39]. DAP and DHXP were detected in different types of face masks at concentrations 1.34 to 8.34 mg/dm^2^, while DPH, DBEP and DINP were not detected [38]. On the other hand, DCHP, DiBP, and DnBP were analyzed in several surgical and N95 masks and only the latter was detected at 0.0018 to 0.22 mg/dm^2^ [40]. Another comprehensive study reports the presence of phthalates in 66 masks from 16 different brands with an average concentration of DBP of 620 ± 497 ng/g, BBP 598 ± 1050 ng/g, DNOP 210 ± 426 ng/g, DEHP 732 ± 1060 ng/g and DEP 276 ± 520 ng/g [39], although this study extracted the phthalates directly from the mask with solvents and did not do a migration study, and thus, results are not comparable. 

The total amount of phthalates quantified in the samples is more than 1000 times lower than the migration limit set for food contact materials by the Regulation 10/2011/Eu of 10 mg/dm^2^, which gives only an indication of their safety for human use, although their use has been associated with inhalation exposure risks [40]. From an environmental point of view, phthalates pose a risk for biota at different trophic levels, according to their predicted no-effect concentration (PNEC) [41]. DEHP has been identified as an endocrine disruptor for which there is scientific evidence of possible adverse effects to the environment [42]. A study performing ecological risk assessment of phthalates concluded that although it is unlikely that DMP and DEP pose risks to marine biota, DBP and DiBP pose low and medium risk for algae and fish, respectively [17]. BBP is classified as a reproductive toxicant and some companies are replacing this product with alternative plasticizers [14]. Similar to MP, the amount of phthalates reaching the water bodies from face masks (considering that 10% of the total masks reach the aquatic environment) is indicated in Figure 14 and ranges from 63 to 187 µg/person per year. Therefore, the phthalate contribution from face masks could pose a risk, considering the spreading use of masks and the toxicological implications they may have. 

Despite not being studied in this work, the release of MP and phthalates in the migration study could imply also a risk for humans, as the use or reuse of masks made from polymeric material can lead to the release of micro and/or nanoplastics [2,43], increasing the risk of MP inhalation through breathing [44,45]. A recent study based on simulated breathing equipment showed that exposure to MP increases as a consequence of erroneous use of masks, while the exposure to MP is reduced when masks are used under their recommended time (<4 h), except when using disinfection pre-treatments [24]. It is therefore paramount to make a proper use of face masks to reduce the risk of MP intake.

## 4. Conclusions

Face masks are mainly made of atactic polypropylene. The migration study herein proposed on face masks is suitable for the evaluation of the release of microplastics as well as phthalates simulating the realistic conditions for which mask components could reach water bodies. Microplastics were released from face masks in the form of fibers and fragments of different sizes, with polypropylene, polyamide, and polyester the main polymers detected. Additionally, phthalates including DBP, BBP, DNOP, and DEHP were also released from face masks. The amounts released were compared to the open bibliography and suggest that improper disposal of face masks can represent an emerging environmental risk considering the huge number of masks being used and directly discharged to receiving waters. It should be noted that cutting the mask in small pieces could represent a bias since it might lead to the release of a higher amount of MP compared to an intact face mask due to a forced fragmentation of the fibers present in the masks’ layers.

The procedure proposed in this study could be applied to investigate the release of face masks components considering also solar radiation and mechanical stress, which could enhance the migration of MP and phthalates and affect the quality of water resources.

## Figures and Tables

**Figure 1 molecules-27-06859-f001:**
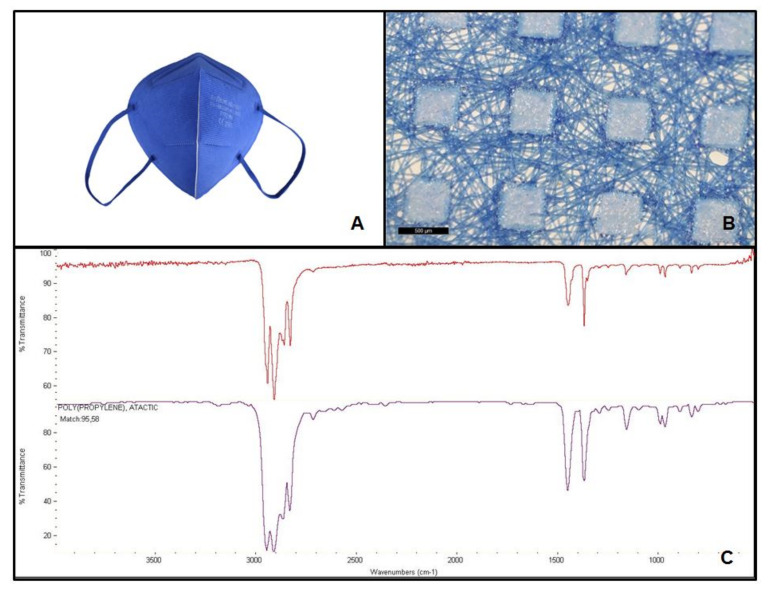
Facial mask FFP2 A studied (**A**); structure after disassembling in layers (500 µm scale bar) (**B**); and mask spectrum (at the top), spectrum listed in the library (at the bottom) identifying polypropylene, atactic, 95.58% (**C**).

**Figure 2 molecules-27-06859-f002:**
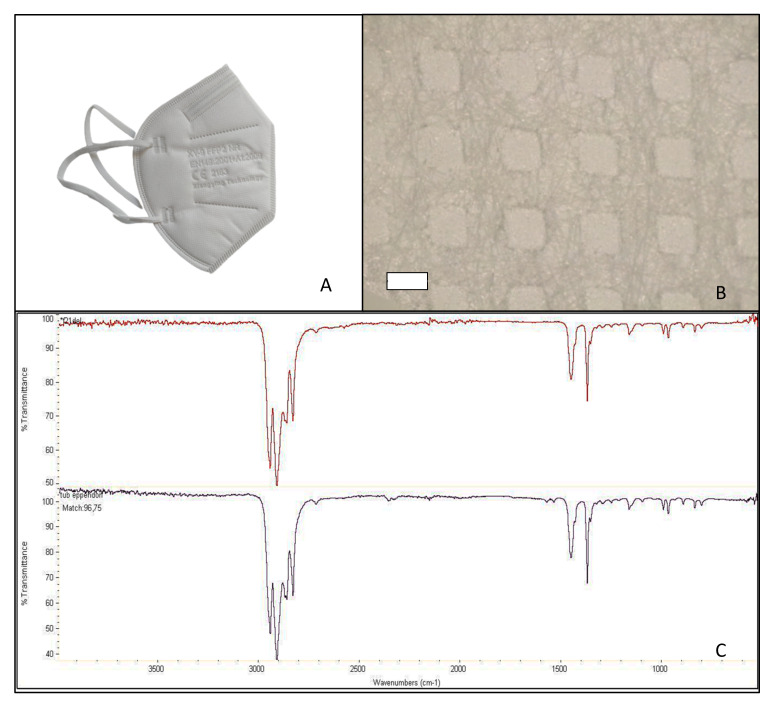
Facial mask FFP2 B studied (**A**); structure after disassembling in layers (500 µm scale bar) (**B**); and mask spectrum (at the top), spectrum listed in the library (at the bottom) identifying polypropylene, atactic, 96.42% (**C**).

**Figure 3 molecules-27-06859-f003:**
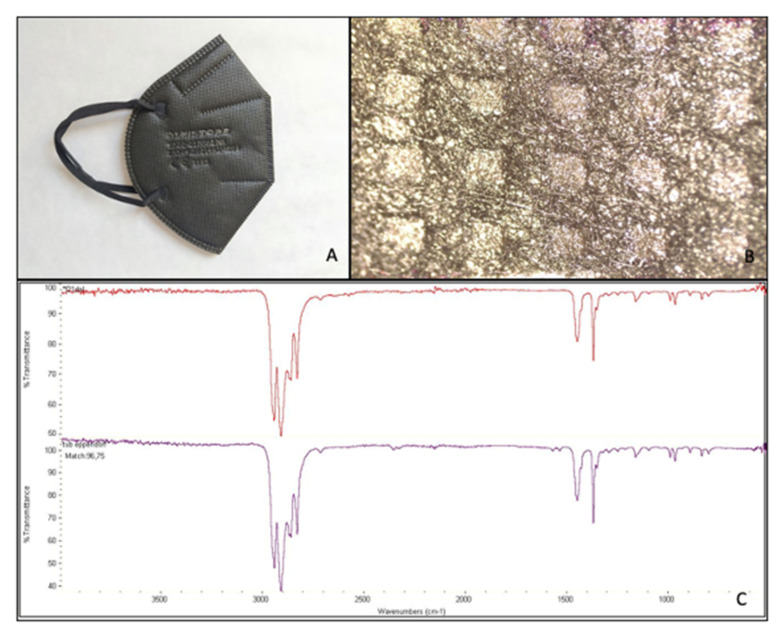
Facial mask FFP2 N studied (**A**); structure after disassembling in layers (500 µm scale bar) (**B**); and mask spectrum (at the top), spectrum listed in the library (at the bottom) identifying polypropylene, atactic, 96.75% (**C**).

**Figure 4 molecules-27-06859-f004:**
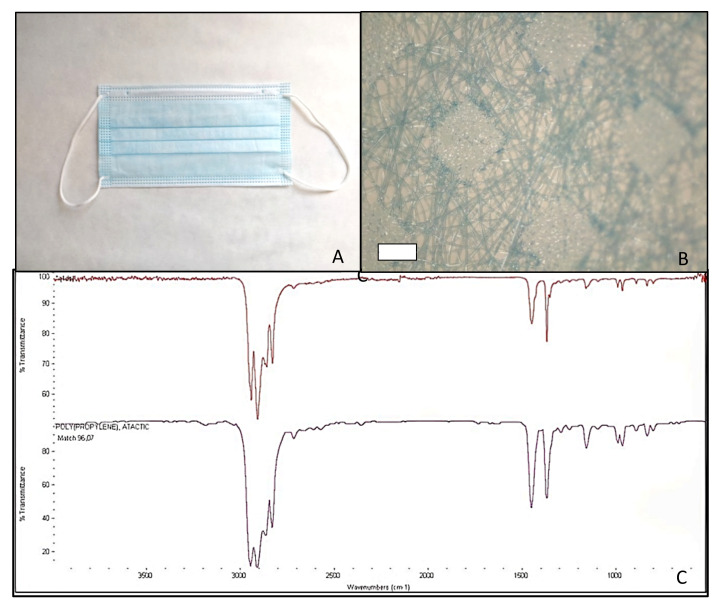
Facial mask Surgical Mask Q studied (**A**); structure after disassembling in layers (500 µm scale bar) (**B**); and mask spectrum (at the top), spectrum listed in the library (at the bottom) identifying polypropylene, atactic, 96.07% (**C**).

**Figure 5 molecules-27-06859-f005:**
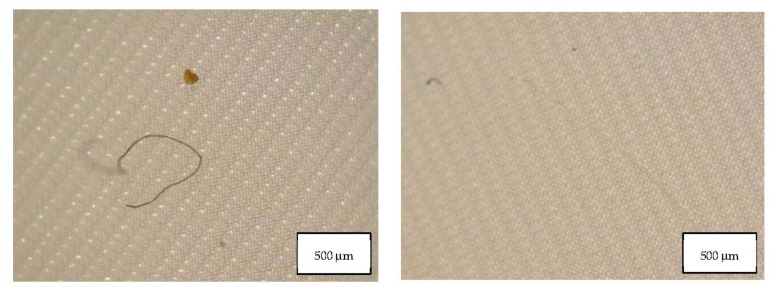
Microplastics detected in the filter of Blank 1 seen at the stereomicroscope with scale bar of 500 µm.

**Figure 6 molecules-27-06859-f006:**
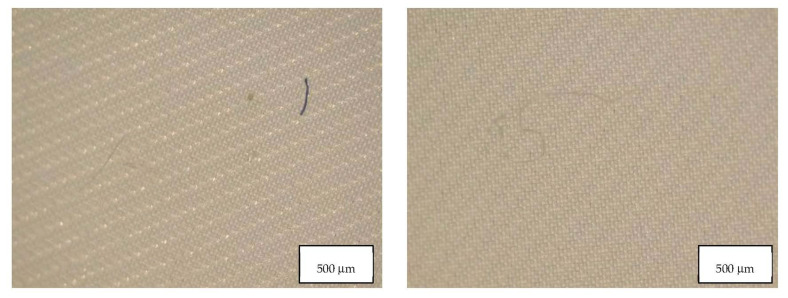
Microplastics detected in the filter of Blank 2 seen at the stereomicroscope with scale bar of 500 µm.

**Figure 7 molecules-27-06859-f007:**
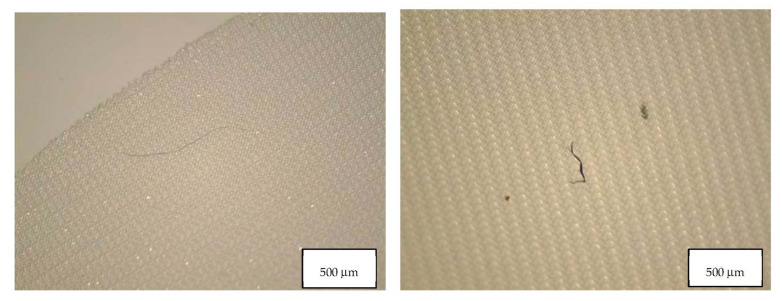
Microplastics detected in the filter of Blank 3 seen at the stereomicroscope with scale bar of 500 µm.

**Figure 8 molecules-27-06859-f008:**
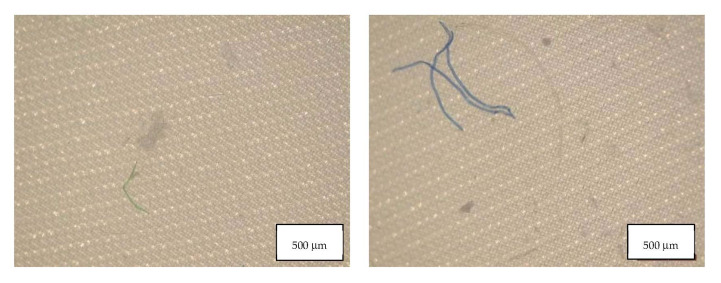
Microplastics detected in the filter of FFP2 A seen at the stereomicroscope with scale bar of 500 µm.

**Figure 9 molecules-27-06859-f009:**
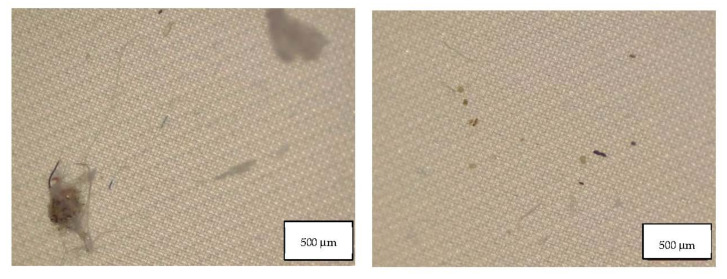
Microplastics detected in the filter of FFP2 B seen at the stereomicroscope with scale bar of 500 µm.

**Figure 10 molecules-27-06859-f010:**
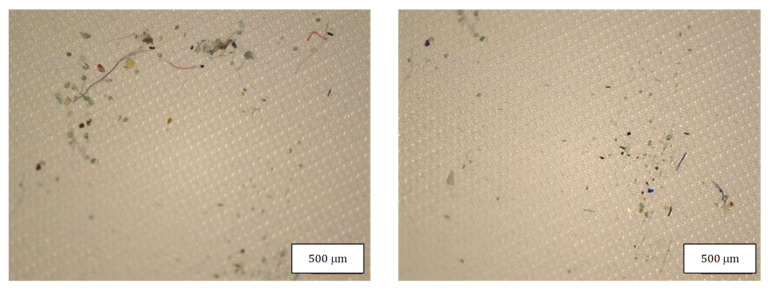
Microplastics detected in the filter of FFP2 N seen at the stereomicroscope with scale bar of 500 µm.

**Figure 11 molecules-27-06859-f011:**
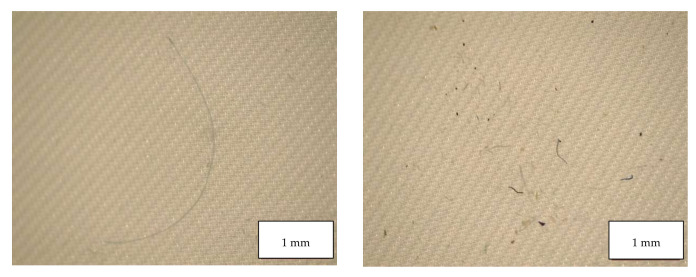
Microplastics detected in the filter of Surgical Mask Q seen at the stereomicroscope with scale bar of 1 mm.

**Figure 12 molecules-27-06859-f012:**
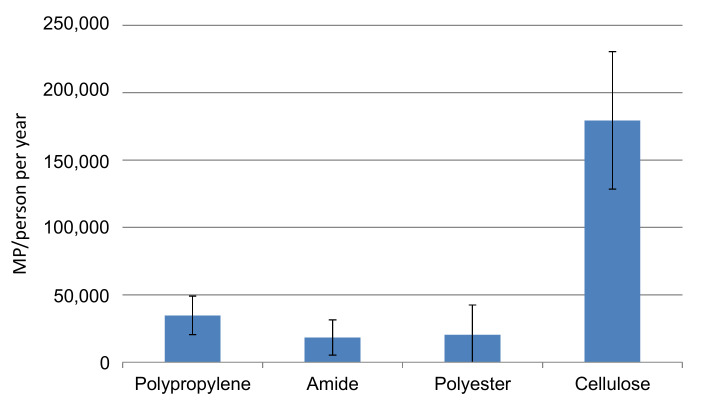
MP released per person/year considering that 10% of face masks used reach the aquatic environment. The data are an estimate based on the results of the current study, considering that each person wears a new mask every 2 days.

**Figure 13 molecules-27-06859-f013:**
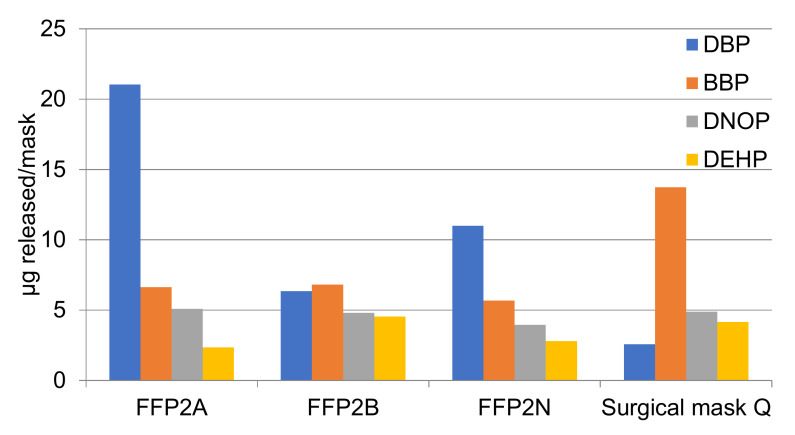
Phthalates migrating from each mask, in µg/mask.

**Figure 14 molecules-27-06859-f014:**
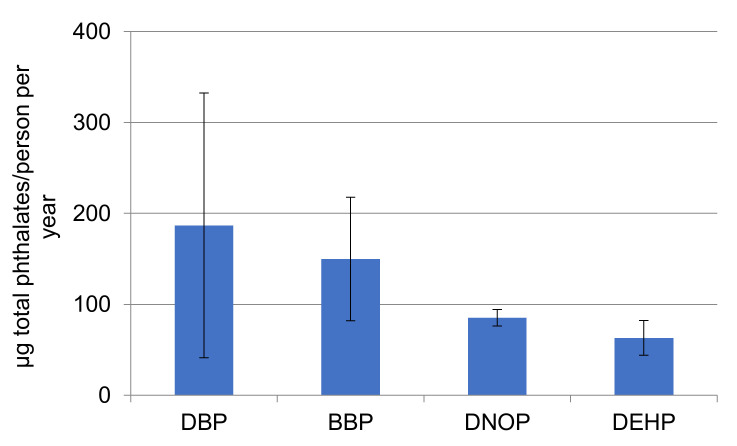
Mean phthalates (*n* = 4) released per person/year considering that 10% of face masks reach the aquatic environment.

**Table 1 molecules-27-06859-t001:** Masks used to perform the migration study, indicating the manufacturer and the model, the Bacterial Filtration Efficiency (BFE), the regulation and the result of the FT-IR analyses expressed as % match with library spectra. Weight (without strings) and area of masks also indicated.

Type of Mask	Manufacturer	Model	BFE (%)	Regulation	% Library Match with PP	Weight (g)	Area (dm^2^)
FFP2 Blue (FFP2 A)	Chang An Da	CAD–01	≥95	EN 149:2001 + A1:2009	95.58	4.286	2.48
FFP2 White (FFP2 B)	Xiang Ying	XY–9	≥94	EN 149:2001 + A1:2009	96.42	4.428	2.55
FFP2 Black (FPP2 N)	1 Mi Store	YJ20–02	≥94	EN 149:2001 + A1:2009	96.75	4.528	2.45
Surgical mask (Q)	Starson	N/A	≥95	Directive 93/42	96.07	3.346	2.62

**Table 2 molecules-27-06859-t002:** Compounds studied ordered by retention time (Rt); CV = Collision Voltage; Q = quantification transition, q = confirmation transition, CE = Collision Energy. Quality parameters of the extraction and analytical method indicating linearity (R^2^ > 0.99), repeatability and reproducibility (%), Instrumental Detection Limits (IDL), recovery (%R), and blank levels obtained in the migration experiments using water as simulant.

Compound	R_t_ (min)	CV (V)	Q (*m*/*z*), CE (eV)	q (*m*/*z*), CE (eV)	Linearity	Repeatability (*n* = 5)	Reprodubility (*n* = 5)	IDL (μg/mL)	%R± RSD (*n* = 5)	Bk Levels (μg/mL)
DMP	3.32	15	195 → 163 (10)	195 → 133 (20)	0.025–1	2.72	20.5	0.00085	92 ± 1.2	ND
DEP	5.05	20	223 → 149 (20)	223 → 177 (10)	0.025–0.75	1.69	2.99	0.00287	91 ± 0.6	0.007
BBP	7.80	15	313 → 91 (10)	313 → 149 (10)	0.05–1	2.30	1.50	0.00002	101 ± 0.6	ND
DBP	7.87	10	279 → 205 (10)	279 → 121 (10)	0.05–1	1.34	2.63	0.00035	112 ± 0.5	2.31
DPP	8.76	35	307 → 149 (20)	307 → 219 (10)	0.025–0.5	3.46	2.45	0.00006	103 ± 0.4	0.6
BMPP	9.15	45	335 → 167 (10)	335 → 251 (10)	0.025–0.75	1.47	3.09	0.00095	90 ± 0.4	0.29
DNHP	9.44	45	335 → 149 (30)	335 → 121 (40)	0.05–1	1.70	2.45	0.00029	81 ± 1.0	ND
HEHP	9.95	15	363 → 149 (20)	363 → 121 (10)	0.025–0.75	4.47	2.19	0.00011	111 ± 0.8	ND
DEHP	10.29	25	391 → 149 (20)	391 → 167 (10)	0.25–1	1.59	1.46	0.00007	83 ± 11	0.001
DNOP	10.46	25	391 → 121 (10)	391 → 261 (10)	0.025–0.75	2.51	1.49	0.00013	97 ± 0.5	0.004
DNP	10.81	15	419 → 149 (20)	419 → 275 (10)	0.05–1	1.39	3.28	0.00003	110 ± 1.3	ND
TPhP d15 (IS)	7.18	25	342 → 262 (20)	342 → 161 (30)	na	na	na	na	na	na

ND = non-detected; na = not analyzed.

**Table 3 molecules-27-06859-t003:** Concentration of MP fragments and fibers in the face masks migration experiment reported as MP/dm^2^ and in number of particles in the blank samples. In parenthesis the percentage of each polymer to ∑MP (including cellulose) detected in each sample is indicated.

Sample Name	Fragments			Fibers		
	PP (%)	PA (%)	Cellulose (%)	PP (%)	PES (%)	Cellulose (%)
FFP2 A	400 (25)	400 (25)	800 (50)	600 (23)	200 (8)	1800 (69)
FFP2 B	200 (25)	0	600 (75)	400 (10)	200 (5)	3600 (85)
FFP2 N	0	600 (21)	2200 (79)	400 (9)	1200 (26)	1500 (65)
Surgical Mask Q	0	600 (33)	1200 (67)	1000 (20)	200 (4)	3800 (76)
Blank 1	0	2 (100)	0	0	0	7 (100)
Blank 2	0	1 (100)	0	0	0	6 (100)
Blank 3	0	0	0	0	1 (17)	5 (83)

**Table 4 molecules-27-06859-t004:** Analysis of particles from the migration experiment (surface area 20 × 0.5 cm^2^) indicating mean length and area of fragments and and length of fibers.

Sample Name	Fragments		Fibers
	Mean Length (μm) ± SD	Mean Area (μm^2^) ± SD	Mean Length (μm) ± SD
FFP2 A	147 ± 69	162 ± 76	210 ± 156
FFP2 B	303 ± 226	259 ±20	339 ± 192
FFP2 N	113 ± 19	115 ± 26	358 ± 231
Surgical Mask Q	209 ± 59	183 ±16	465 ± 342
Blanks ^1^	114 ± 0.4	128 ± 8	1217 ± 947

^1^ Mean of 3 blanks.

**Table 5 molecules-27-06859-t005:** Number of MP released from face masks according in this study and comparison to previous studies.

Sample	MP Released per Mask	Type of Study	Study Length	Detection Method	Reference
FFP2 A	3968 ^1^	Migration study in water	24 h	µ-FT-IR	Present study
FFP2 B	2040 ^1^				
FFP2 N	5390 ^1^				
Surgical Mask	4716 ^1^				
N95	110	Breathing simulation	24 h	Laser Infrared Imaging system	Li et al., 2021 [24]
Surgical Mask 1	137				
Surgical Mask 2	264				
Cotton	222				
Fashion	185				
Non-woven	150				
Activated carbon	540				
N95 mask 1	4400				
N95 mask 2	3700				
Surgical Mask 1	1800				
Surgical Mask 2	1600				
Surgical Mask 3	1500				
Surgical Mask 4	1300	Migration study	30 min	FT-IR	Ma et al., 2021 [10]
Surgical Mask 5	1700				
Surgical Mask 6	1700				
Surgical Mask 7	1700				
Surgical Mask 8	2900				
Surgical Mask (*n* = 16)	28,000 ± 5000	Shear stress	120 s	Flow cytometry	Morgana et al., 2021 [9]
Surgical Mask 1	78,000	Migration study in seawater	180 h UV irradiation, 24 h stirring into artificial seawater	µ-FT-IR	Saliu et al., 2021 [25]
Surgical Mask 2	55,000				
Surgical Mask 3	173,000				
Surgical Mask 4	110,000				
Surgical Mask 5	95,000				
Surgical Mask 6	67,000				
Surgical Mask 7	142,000				
Surgical Mask reused	3600	Migration study in water	24 h	µ-FT-IR	Shen et al., 2021 [23]
Surgical Mask reused	5400	Migration study in detergent			
Surgical Mask reused	4400	Migration study in alcohol			

^1^ In the present study, the amount of MP released referred to the MP released from 20 × 0.5 cm^2^ of masks. Therefore, these numbers have been obtained by relating the results to the whole surface of face masks (Table 1 for mask dimensions).

**Table 6 molecules-27-06859-t006:** Phthalate concentration in the present study and comparison with other studies on surgical, N95 or activated charcoal (AC) masks in mg/dm^2^. ND = Not Detected; - = phthalate not included in the study. The blank contribution has been already subtracted from the phthalate concentration.

Sample	Phthalate Concentration mg/dm^2^								
	DBP	BBP	DNOP	DEHP	DEP	DPP	DMP	Technique	Reference
FFP2 A	0.0085	0.0027	0.0020	0.0009	ND	ND	ND	UPLC-MS/MS	Present Study
FFP2 B	0.0025	0.0027	0.0019	0.0018	ND	ND	ND		
FFP2 N	0.0045	0.0023	0.0016	0.0011	ND	ND	ND		
Surgical Mask Q	0.0010	0.0052	0.0019	0.0016	ND	ND	ND		
Surgical 1 ^1^	1.18 ± 0.05	1.72 ± 0.10	ND	2.91 ± 0.07	ND	0.87 ± 0.02	ND	in-situ DCBI-MS/MS	Min et al., 2021 [38]
Surgical 2 ^1^	1.56 ± 0.03	1.53 ± 0.05	ND	3.74 ± 0.14	ND	3.02 ± 0.12	ND		
Surgical 3 ^1^	3.20 ± 0.10	2.00 ± 0.09	ND	3.29 ± 0.16	ND	1.31 ± 0.08	ND		
N95 1 ^1^	ND	1.76 ± 0.09	ND	1.28 ± 0.06	ND	1.14 ± 0.07	ND		
N95 2 ^1^	0.68 ± 0.04	2.76 ± 0.12	ND	1.55 ± 0.09	ND	0.81 ± 0.04	ND		
AC 1 ^1^	ND	1.93 ± 0.05	ND	0.84 ± 0.01	ND	2.30 ± 0.06	ND		
AC 2 ^1^	ND	1.77 ± 0.03	ND	0.71 ± 0.01	ND	1.63 ± 0.05	ND		
Surgical 1	-	ND	ND	0.028	ND	-	0.0005	Pyr-CG/MS	Wang et al., 2022 [40]
Surgical 2	-	ND	ND	0.020	0.0005	-	0.0001		
Surgical 3	-	ND	ND	ND	ND	-	0.0005		
Surgical 4	-	ND	ND	ND	0.0027	-	0.0004		
Surgical 5	-	ND	ND	ND	0.0016	-	0.0003		
Surgical 6	-	ND	ND	ND	0.0077	-	0.0001		
Surgical 7	-	ND	ND	ND	0.0062	-	0.0001		
Surgical 8	-	ND	ND	ND	0.0011	-	ND		
Surgical 9	-	ND	ND	ND	0.0013	-	0.0001		
Surgical 10	-	ND	ND	0.019	ND	-	0.0001		
Surgical 11	-	ND	ND	ND	0.0030	-	0.0001		
Surgical 12	-	ND	ND	ND	ND	-	0.0003		
N95 1	-	ND	ND	0.057	0.0082	-	ND		
N95 2	-	ND	ND	0.12	0.0061	-	0.0002		
N95 3	-	ND	ND	0.088	ND	-	0.0001		
N95 4	-	ND	ND	0.039	0.0029	-	0.0010		

^1^ Different brands, for each mask *n* = 3. ND not detected.

## Data Availability

The data presented in this study and the Appendix A are available on request from the corresponding author.

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
