# Peer review of "Migration of Microplastics and Phthalates from Face Masks to Water"

_molecules, 2022, doi:10.3390/molecules27206859_

Round 1

Reviewer 1 Report

The Authors presented the results of possible sides effect of using face masks for the environment. First, I was afraid of the suitability of the manuscript for the aims and scopes for Molecules, but... in my opinion, the results can be presented in Journal. Especially, the results show the emerging problem with the possible pollution from the facemasks, and possible contamination molecules release into the water, and then to living organisms. So, I m not worried about the suitability. Nevertheless, the manuscripts need major revision before publication. Detailed comments are listed below:

Main comments:

- FTIR spectra of all types of masks should be included in the main text

- consider adding more information from supporting materials into the main text. Honestly, all of the figures are suitable here, and could strongly improve manuscript quality

- include images of the microplastics in the main text.

- please highlight why did you choose these kinds of masks (why not fpp3, etc. Some additional comments, like the dependence of the manufacturer, might be also considered here.

line numbering is helpful during the review.

Generally, the introduction is well organized and allows readers to follow up on the idea of the manuscripts. Minor updates:

- Please add unit 1300 to 4400 (page 2)

- Please add comments on what method is recommended for face mask utilization.

Materials and methods

- please check the manuscript content before submission i.e. page 2 Error! Reference source not found.).

- just a short question - did you consider the impact of cutting (which damaged the masks) on the presented results? The structure is not uniform which might be caused additional plastics release. You can add some comments. On the other hand, during the wearing damage process, i.e. caused by rubbing might occur.

 p. 2.4 what do you mean by 64 scans. Did you perform mapping?

Concentrations are given in MP/dm2 - usually for concentration is dm3, but probably you mean here appearance - just to avoid some misunderstandings please add some short comments.

Results and discussion 

- please check the manuscript before the final presentation (see Error!)

- figure 4 please add some additional comments on why they're so wide ranges

- page 8 - see 1st sentence there is "." Is that missing something or just a type error?

- section 3 should be named results and discussion

Author Response

Review Report form

We thank the reviewers for the time dedicated to review the manuscript and for the constructive comments. Our responses to the specific comments are given in italics.

Reviewer 1

The Authors presented the results of possible sides effect of using face masks for the environment. First, I was afraid of the suitability of the manuscript for the aims and scopes for Molecules, but... in my opinion, the results can be presented in Journal. Especially, the results show the emerging problem with the possible pollution from the facemasks, and possible contamination molecules release into the water, and then to living organisms. So, I am not worried about the suitability. Nevertheless, the manuscripts need major revision before publication. Detailed comments are listed below:

Main comments:

  1. FTIR spectra of all types of masks should be included in the main text

Response: FTIR spectra were added in the main text.

  1. Consider adding more information from supporting materials into the main text. Honestly, all of the figures are suitable here, and could strongly improve manuscript quality

Response: Supporting material is included in the manuscript

  1. Include images of the microplastics in the main text.

Response: Two images from fragments and fibers released from FFP2 N were included.

  1. Please highlight why did you choose these kinds of masks (why not fpp3, etc. Some additional comments, like the dependence of the manufacturer, might be also considered here.

Response: Surgical and FFP2 masks were used as representative samples since they were the recommended masks to wear and the most widely used during the pandemic. We tested the masks that were provided to IDAEA-CSIC personnel. This information has been added.

  1. Line numbering is helpful during the review.

Response: Line numbers have been added.

Generally, the introduction is well organized and allows readers to follow up on the idea of the manuscripts. Minor updates:

  1. Please add unit 1300 to 4400 (page 2)

Response: Comment addressed

  1. Please add comments on what method is recommended for face mask utilization.

Response: recommendation on proper use of face masks has been included in the conclusions.

Materials and methods

  1. please check the manuscript content before submission i.e. page 2 Error! Reference source not found.).

Response: Corrected

  1. Just a short question - did you consider the impact of cutting (which damaged the masks) on the presented results? The structure is not uniform which might be caused additional plastics release. You can add some comments. On the other hand, during the wearing damage process, i.e. caused by rubbing might occur.

Response. No, the impact of cutting was not taken into account although the cutting process would damage the edge of the mask and subsequently the sample. We included a comment in the conclusions explaining that cutting the face masks could bias the results since additional release of microplastics could have taken place.

  1.  p. 2.4 what do you mean by 64 scans. Did you perform mapping?

Response: No mapping was performed, for the analysis 64 scans were done in order to accumulate multiple scans which improve the quality of the spectra.

  1. Concentrations are given in MP/dm2 - usually for concentration is dm3, but probably you mean here appearance - just to avoid some misunderstandings please add some short comments.

Response: The concentrations are given in MP/dm2 as the first data were related to the 0.5 cm2 piece of masks used to perform the migration test. The dm2 unit was used as it is the units indicated in Regulation 10/2011. Then, the results were converted into MP per mask in order to be able to compare the results obtained from other studies.

Results and discussion 

  1. please check the manuscript before the final presentation (see Error!)

Response: Corrected

  1. Figure 4 please add some additional comments on why they're so wide ranges

Response: Figure 4 is now Figure 15. The following comment was added: The masks taken into account in the present study were purchased from different manufacturers and therefore their composition differed in terms of plasticizers used leading to high standard deviation in the mean mg of phthalates released per person per year

  1. page 8 - see 1st sentence there is "." Is that missing something or just a type error?

Response: Corrected

  1. Section 3 should be named results and discussion

Response: Corrected

Reviewer 2 Report

This study describes the migration of MP and some plasticizer compounds from different types of face masks. The article has been well organized and deservs to be considering to be published in molecules after minor corrections. 

In order to improve the quality of the article some I made some comments in the attached manuscript. 

Author Response

Reviewer 2

Gave the comments directly on a pdf file and they have been addressed in the version of the .doc manuscript sent.

Response: Corrected, changes in the word document. Thanks for the comments. 

We were asked to provide a reference for the phthalate extraction and the chromatographic condition. However, the method was developed in this study for face masks, and we provide the protocol for the extraction and the chromatography and mass spectra parameters.

Reviewer 3 Report

It remains open how the authors calculated the content of MP per dm² of mask or per mask. Spectroscopic methods are inadequate for mass calculations compared to thermo-analytical  methods. Provide mass-based data for the risk assessment - urgently needed. The topic is interesting.

Author Response

It remains open how the authors calculated the content of MP per dm² of mask or per mask

Response: In order to calculate the content of MP/dm2 a proportion was applied. In fact, the amount of MP released and presented in Table 3 refer to the MP released from 20 pieces of 0.5 cm2. The note in Table 6 states that the amount of MP released per mask have been obtained by relating the amount of MP in Table 5 to the whole surface of the mask.

Spectroscopic methods are inadequate for mass calculations compared to thermo-analytical methods. Provide mass-based data for the risk assessment - urgently needed. The topic is interesting.

The spectroscopic method (FTIR) are used to identify the polymer type in masks form the migration experiments. The weight of those polymers is impossible to weigh. Therefore data is provided as number pt particles of each polymer/dm2.Then this data was concerted to MP/mask, to estimate the number of particles release to water considering that a mask was discharged to waters. The risk this may cause cannot be assessed in the present study which was more aimed to determine the method to estimate the release of MP and phthalates from face masks to waters. Further studies will be done on that direction.

Round 2

Reviewer 1 Report

The Authors improved the manuscript due to my recommendations.

The manuscript can be published in its present form.

The figures included in the main text strongly improved manuscript quality.

Thank you for answering my questions about the sample preparation.

Reviewer 3 Report

A thermo-analytical method of the water sample could improve the knowledge about the release potential. I see the paper as a first approximation and information about the release potential in combination with phthalates into the water phase and the air during breathing. A non-target screening for further compounds would improve the risk assessment of masks.